# TrGa: Reconsidering the Application of Graph Neural Networks in Two-View Correspondence Pruning

Luanyuan Dai
Nanjing University of Science and Technology
Nanjing, China
dailuanyuan@njust.edu.cn

Xiaoyu Du
Nanjing University of Science and Technology
Nanjing, China
duxy@njust.edu.cn

Jinhui Tang*
Nanjing University of Science and Technology
Nanjing, China
jinhuitang@njust.edu.cn

## ABSTRACT

Two-view correspondence pruning aims to accurately remove incorrect correspondences (outliers) from initial ones. Graph Neural Networks (GNNs) incorporated by Multilayer Perceptrons (MLPs) are treated as a powerful manner to handle sparse and unevenly distributed data. However, the expression capability of correspondence features obtained by MLPs is limited by their inherent insufficient of context information. In addition, previous works directly utilize the outputs of off-the-shelf GNNs, thus leading to confusion between sparse correspondence attribute features and their global structural information. To alleviate these issues, we propose a two-view correspondence pruning network TrGa. Specifically, we firstly use complete Transformer structures instead of context-agnostic MLPs to capture correspondence features with global context information and stronger expression capability. After that, we introduce the Concatenation Graph Node and Global Structure (CGNS) block to separately capture the interaction patterns among sparse correspondence attribute features and the global structural information among them, which can prevent their confusion. Finally, the proposed Feature Dimension Transformation and Enhancement (FDTE) block is applied for dimension transformation and feature augmentation. Additionally, we propose an efficient variant C-TrGa, in which the similarity matrix of the proposed C-Transformer is computed along the channel dimension. Extensive experiments demonstrate that the proposed TrGa and C-TrGa outperform state-of-the-art methods in different computer vision tasks.

## CCS CONCEPTS

• **Computing methodologies** → **Matching**; • **Information systems** → *Similarity measures*; Combination, fusion and federated search.

## KEYWORDS

Correspondence Pruning, Attribute Feature, Global Structural Information, Graph Neural Network, Transformer

*Corresponding author.

**ACM Reference Format:**
Luanyuan Dai, Xiaoyu Du, and Jinhui Tang. 2024. TrGa: Reconsidering the Application of Graph Neural Networks in Two-View Correspondence Pruning. In *Proceedings of the 32nd ACM International Conference on Multimedia (MM '24), October 28-November 1, 2024, Melbourne, VIC, Australia.* ACM, New York, NY, USA, 10 pages. https://doi.org/10.1145/3664647.3681139

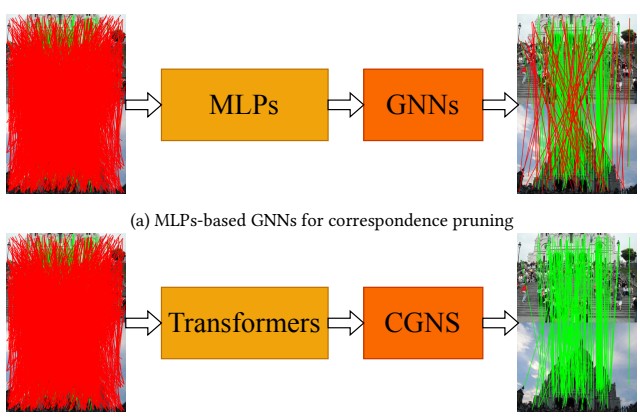

(a) MLPs-based GNNs for correspondence pruning

(b) Illustration of the proposed TrGa

**Figure 1: GNNs-based correspondence pruning networks with different architectures. (a) is the unified architecture of other MLPs-based GNNs for correspondence pruning. (b) illustrates the architecture of the proposed TrGa, in which Transformers are used to extract correspondence features with strong expressive capabilities, followed by the proposed CGNS operation to construct a robust global graph. The green and red lines show inliers and outliers, respectively.**

## 1 INTRODUCTION

Good-quality correspondences are crucial for many important computer vision tasks, *e.g.*, visual localization [39, 40], image fusion [30, 44, 50], image registration [23, 25], Simultaneous Location and Mapping (SLAM) [33], point cloud registration [3, 31], Structure from Motion (SfM) [37, 41], etc. A popular approach is to construct a putative correspondence set based on local feature matchings (SIFT [27], SuperPoint [12], etc.). But unfortunately, the putative correspondence set is polluted by massive incorrect correspondences (i.e., outliers), as shown on the left of Figure 1, due to passive influence of rotations, illumination changes and viewpoint changes, etc. To reduce the negative impact of outliers, it is common to remove

outliers and select a reliable subset composed as much as possible of true correspondences (i.e., inliers).

Recently, owing to the powerful data processing capabilities of Graph Neural Networks (GNNs), previous works (OA-Net++ [55], LMC-Net [24], CL-Net [58], MS$^2$DG-Net [11], U-Match [21], MGNet [28] and NCMNet [22]) have endeavored to integrate GNNs with MLPs-based backbones to remove outliers from sparsely and unevenly distributed correspondences (correspondence pruning), achieving notable success. But they generally choose the off-the-shelf GNNs (*e.g.*, graph convolutional network (GCN) [18]), which utilize the ready-made formulation $F_{gcn} = L_{sym}F^T\tilde{A}$ to simultaneously learn and apply both the attribute information and global structural information of sparse correspondences. Therefore, during the information propagation process, the attribute information and global structural information of sparse correspondences mutually influence and intertwine, blurring the boundaries between them and even consequently causing confusion, which can impact correspondence pruning network performance. Another issue is that current works employ Multilayer Perceptrons (MLPs) as the backbone which cannot capture sufficient context information, resulting in inadequate expressive power of the obtained correspondence features. This makes it difficult to construct robust global graphs, leading to a decrease in network performance. (See in Table 10.) More specifically, all of them first employ MLPs to extract correspondence features, followed by various strategies to construct graphs to capture global or local information among sparse correspondences. Some works [21, 22] attempt to extract features by MLPs at different granularities, but the parameter quantity will be greatly increased. Others [22, 58] try to gradually remove outliers to increase operational efficiency, which may reduce the ability of network to resist noise interference. However, due to the insufficient of context information in correspondence features extracted by MLPs, the constructed global graph is not sufficiently robust, which significantly reduces the effectiveness of the model, as shown in Figure 1 (a).

To tackle the above issues, we reconsider the application of GNNs in correspondence pruning instead of directly utilizing existing GNNs. The attribute information of sparse correspondences is the intrinsic characteristics of sparse correspondences. Its incorporation enables the model to better understand the relationships among sparse correspondences, thereby improving model performance. Additionally, global structural information contains knowledge that is not present in attribute information, which plays a crucial role in characterizing correspondences and representing relationships among correspondences in each image pair. Therefore, how to effectively use both types of information becomes paramount. Here, we propose TrGa for two-view correspondence pruning, which is made of a Correspondence Feature Extractor (CFE), a Global Graph Construction (GGC), and a Prediction Block, as shown in Figure 2. CFE is used to extract correspondence features, which comprise five complete Transformer structures instead of MLPs. This is because Transformers can naturally capture global context information, whereas context-agnostic MLPs cannot. GGC includes a Concatenation Graph Node and Global Structure (CGNS) block and a Feature Dimension Transformation and Enhancement (FDTE)

block. The proposed CGNS is employed to better explore complementary relationships between attribute information and global structural information of sparse correspondences. As shown in Figure 3, the CGNS block is used to separately capture the interaction patterns among sparse correspondence attribute features and the global structural information among sparse correspondences. Instead of directly multiplying them as in GCN, they are concatenated together, which prevents confusion between the two types of information, thereby enhancing network effectiveness. Furthermore, we also demonstrate that the CGNS block is equivalent to a low-pass filtering operation, and low-frequency signals contain more effective information [34]. Then, we use the FDTE block for dimension transformation and feature enhancement. Finally, we employ the prediction block for probability prediction, judging the correctness of correspondences based on probabilities and conducting subsequent experiments. In addition, we propose an efficient variant C-TrGa, in which the similarity matrix of the proposed C-Transformer is computed along the channel dimension.

Our contributions are as follows: 1) In this work, we use complete Transformer structures or efficient C-Transformers instead of MLPs to extract correspondence features, which can overcome the individual feature-extraction limitation of MLPs. 2) The proposed CGNS block can separately capture the interaction patterns among sparse correspondence attribute features and the global structural information among sparse correspondences, to prevent their confusion. 3) A simple and effective network TrGa and its faster variant C-TrGa, are proposed for correspondence pruning, which perform best on relative pose estimation, homography estimation, visual localization and point cloud registration tasks with acceptable parameter quantities.

## 2 RELATED WORK

### 2.1 Learning-Based Correspondence Pruning.

RANSAC [14] and its variants (MAGSAC [6], MLESAC [46] and so on) fail in existing datasets with a high proportion of outliers, so deep learning-based correspondence pruning networks have emerged and made breakthrough progress. Specifically, CNe [32] is a pioneering work that demonstrates it is feasible to directly use correspondence coordinates for correspondence pruning. Inspired by attention mechanism [47], ACNe [43] and LAGA-Net [10] pay more attention to important correspondences and reduces attention to outliers, to improve network performance. After that, ANA-Net [52] proposes the concept of attention in attention, which essentially explores the similarity between attention weights. Hence, attention in attention can also be explained as the second-order attention. NM-Net [57] uses affine attributes to find compatibility-specific neighbors to aggregate features. ConvMatch [56] forcibly places unordered sparse correspondences into a dense motion field and processes them by CNNs. In addition, GNNs have also been used in correspondence pruning in recent years, which will be introduced in Section 2.2.

### 2.2 GNNs in Correspondence Pruning.

Graph Neural Networks (GNNs) have powerful abilities to discover and extract features in graph structured data, so they have been used in correspondence pruning. Inspired by DIFFPOOL [53], OA-Net++

[55] adopts the differentiable pooling and unpooling operations to cluster sparse correspondences. LMC-Net [24] uses GNNs to mine the motion coherence property among sparse correspondences. In CL-Net [58], Zhao et al. perform progressive pruning on initial correspondences according to consensus scores obtained from local-to-global dynamic graphs to reduce the negative impact of outliers. MS$^2$DG-Net [11] borrows attentions from Transformer [47] and combine them with GNNs to improve the ability of network to extract and aggregate information. In U-Match [21], Li et al. imitate the U-shaped network to construct graphs at different granularities, which can combine information of different granularities, to improve network performance. In NCMNet [22], Liu et al. use a plain graph convolutional network (GCN) [18] to establish a global graph space to find more accurate neighbors.

## 2.3  Transformers in Vision Related Tasks.

ViT [13] tries to use a standard Transformer [47] to complete computer vision tasks, which demonstrates that the gap between Natural Language Processing (NLP) and Computer Vision (CV) can be broken. Inspired by this work, Transformers have achieved great success in various computer vision tasks, *e.g.*, object detection, image restoration, semantic segmentation and so on. DETR [7] is an end-to-end object detection network, in which Nicolas et al. utilize the global modeling ability of Transformers to achieve no redundancy boxes and success. After that, Liu et al. [26] select shifted windows to capture multiple scale information, so that the problem of detecting objects of different sizes in a single image has been solved. This idea is used in SwinFusion [29]. Transformer is used in zero-shot anomaly detection [60]. Segmenter [42] extends Transformers to semantic segmentation, in which the proposed network models global context throughout the network. At the same time, Transformers have been used in the vision-and-language field and obtained competitive performance results with significantly less network running cost in ViLT [17].

## 3  PROPOSED METHOD

### 3.1  Problem Formulation.

We first use local features (SIFT [27], SuperPoint [12] and so on.) and a nearest neighbor matching strategy to build a putative over-complete correspondence set $C$ for an image pair $(I^s, I^t)$:

$$C = \{c_1; c_2; ...; c_N\} \in \mathbb{R}^{N \times 4}, c_i = \left(x_i^s, y_i^s, x_i^t, y_i^t\right), \quad (1)$$

where $c_i$ is a correspondence between normalized interesting points $\left(x_i^s, y_i^s\right)$ and $\left(x_i^t, y_i^t\right)$ in the given image pair.

The proposed TrGa is employed to obtain the final probability set $P = \{p_1; p_2; ...; p_N\}$, in which $p_i \in [0, 1)$ shows the inlier probability of the $i^{th}$ correspondence. The above operations can be written as:

$$Z = f_\psi(C), \quad (2)$$

$$P = Pre(Z), \quad (3)$$

where $f_\psi(\cdot)$ presents TrGa or C-TrGa with their parameters $\psi$; $Pre(\cdot)$ describes a prediction block; $Z$ is the logit value set for classification.

### 3.2  Main Structure of TrGa.

The proposed TrGa (shown in Figure 2) consists of a Correspondence Feature Extractor (CFE) , a Global Graph Construction (GGC), and a Prediction Block, which is very simple and efficient. $Up(\cdot)$ layer is used to project the initial input correspondence set $C \in \mathbb{R}^{4 \times N}$ into a correspondence feature set $F = \{f_i\}_{i=1}^N \in \mathbb{R}^{S \times N}$. Next, CFE is used to further extract a stronger expressive power feature set $F_T \in \mathbb{R}^{S \times N}$ from the correspondence feature set $F$, due to Transformer strong global modeling ability. After that, GGC is employed to capture attribute information and global structural information effectively from the stronger feature set $F_T$ without a confusion, both of which are important information contained in graph data. Notably, attribute information denotes the inherent properties of sparse correspondences, while global structural information describes the potential relationships among them in each image pair. (We will explain them in Section 3.4.) Finally, we obtain the final probability set $P$ by the prediction block.

### 3.3  Correspondence Feature Extractor.

As shown in Figure 2, the Correspondence Feature Extractor (CFE) consists of one $Up(\cdot)$ layer and five Transformer blocks. The $Up(\cdot)$ layer is used to transform low-dimensional correspondences into high-dimensional features for better feature extraction. Here, we choose a simple MLP layer as the $Up(\cdot)$ layer. And Transformer blocks are used to capture correspondence features with stronger representation capabilities, each of which is made of two PreNorms (PNs), one Multi-Head Self-Attention (MHSA) and a FeedForward (FF), as shown in Figure 2. Layer normalization [2] is selected as a PreNorm (PN), which is good at handling variable length data. Following ViT [13], a FeedForward (FF) consists of two linear layers and a GELU layer [15]. Specifically, the correspondence feature set $F = \{f_i\}_{i=1}^N \in \mathbb{R}^{S \times N}$ passes through a PN layer and a MHSA layer, and a residual structure is used. The output result is put into a PN layer and a FF layer, followed by a residual structure. These operations can be written as:

$$\tilde{F}_{PM} = MHSA\left(PN\left(F\right)\right) + F, \quad (4)$$

$$\tilde{F}_{PF} = FF\left(PN\left(\tilde{F}_{PM}\right)\right) + \tilde{F}_{PM}. \quad (5)$$

More specifically, the correspondence feature set $F$ is linearly transformed into a query set $Q$, a key set $K$ and a value set $V$, respectively. The self-attention (SA) is applied on $Q$, $K$ and $V$ by Eq. (6), as follows:

$$SA\left(F\right) = softmax\left(Q \times K^\mathsf{T}\right) \times V, \quad (6)$$

where the similarity matrix $A_s = softmax\left(Q \times K^\mathsf{T}\right)$ is to obtain similarity among correspondences; $\times$ and $Softmax(\cdot)$ present a matrix multiplication operation and a softmax operation.

Notably, the MHSA is an extension of the SA, where $H$ self-attention operations are performed in parallel, named "heads", and their outputs are concatenated, as follows:

$$MHSA\left(F\right) = U_{out}\left(\left[SA_1\left(F\right) || \cdots || SA_H\left(F\right)\right]\right), \quad (7)$$

where $U_{out}$ is created by one linear projection; $[\cdot || \cdot]$ presents the concatenation operation.

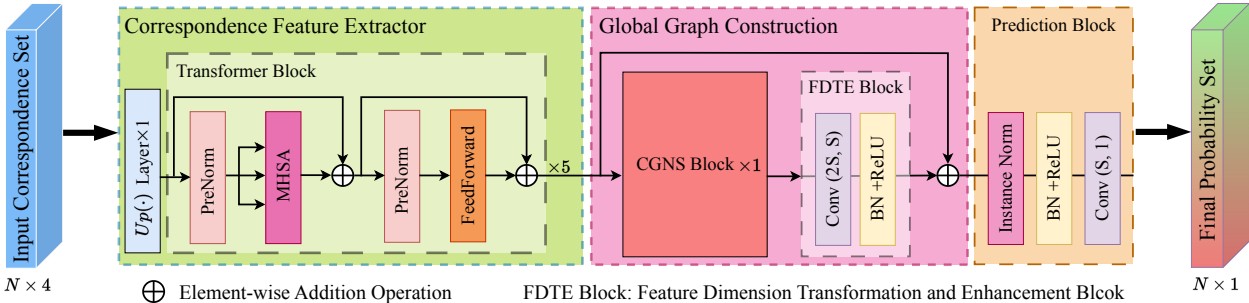

**Figure 2: The architecture of TrGa is composed of a Correspondence Feature Extractor (CFE) , a Global Graph Construction (GGC), and a Prediction Block.**

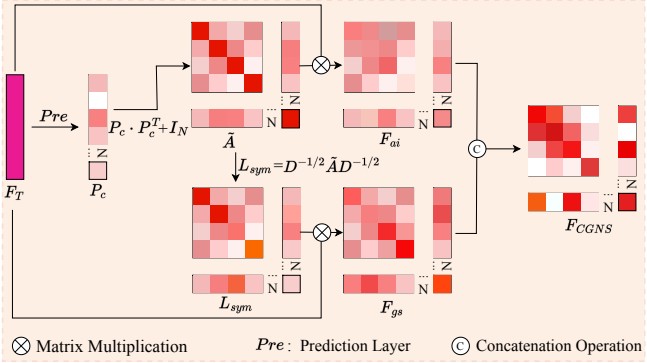

**Figure 3: The architecture of the proposed Concatenation Graph Node and Global Structure (CGNS) Block , in which the color of the block is closer to red, the value is larger.**

Although the standard Transformer has achieved good results, the theoretical time complexity is too high, due to the fact that the similarity matrix $A_s \in \mathbb{R}^{N \times N}$, and $N$ is up to 2000. Therefore, we propose another version similarity matrix $A_c \in \mathbb{R}^{S \times S}$ along the channel dimension, where $S = 128$. (See **Efficiency** in Section 4.5.)

### 3.4 Global Graph Construction.

The more expressive features $F_T$ obtained by CFE is used to construct global graph. As shown in Figure 2, Global Graph Construction (GGC) consists of a Concatenation Graph Node and Global Structure (CGNS) block and a Feature Dimension Transformation and Enhancement (FDTE) block, which will be introduced in turn.
**CGNS Block.** The CGNS block is used to separately capture the interaction patterns among sparse correspondence attribute features and their global structural information to prevent confusion between attribute information of sparse correspondences and the global structural information among them. Attribute information refers to the intrinsic characteristics of sparse correspondences, while global structural information delineates the potential relationships among them in each image pair.

As shown in Figure 3, $F_T$ is firstly passed through a prediction block, so that we can obtain a sparse correspondence probability set $P_c$. After that, we build Adjacent Matrix $A \in \mathbb{R}^{N \times N}$ to explore

relationships in every two members in the $P_c$. Unfortunately, Adjacent Matrix $A \in \mathbb{R}^{N \times N}$ cannot consider its own information, so a self-loop is created on top of it to consider its own information and keep numerical stability. These can be denoted as:

$$A = P_c \cdot P_c^{\mathsf{T}}, \tag{8}$$

$$\tilde{A} = A + I_N, \tag{9}$$

where $\tilde{A} = \left\{ \tilde{A}_{i,j} \right\}_{i,j=1}^{N} \in \mathbb{R}^{N \times N}$ is the final Adjacent Matrix and $I_N$ is an $N \times N$ unit matrix.

After that, we embed the final Adjacent Matrix $\tilde{A}$ (attribute information) into sparse correspondences, which can learn interaction patterns among sparse correspondence attribute features and can be written as:

$$F_{ai} = F_T \times \tilde{A}. \tag{10}$$

Next, we use the normalized Laplace matrix $L_{sym}$ to aggregate neighbor structural information from a spatial perspective. In addition, the proposed CGNS block operates on all sparse correspondences in each given image pair at once in GGC. Hence, this process can be considered as learning global structural information among sparse correspondences and can be denoted as:

$$F_{gs} = L_{sym} \times (F_T)^{\mathsf{T}}, \tag{11}$$

where $L_{sym} = D^{-1/2}\tilde{A}D^{-1/2}$ [34][1], is the normalized Laplace matrix. $D$ is the diagonal degree matrix of $\tilde{A}$.

Finally, we use interaction patterns among sparse correspondence attribute features and global structural information among them at the same time, which can be written as:

$$F_{CGNS} = [F_{ai}||F_{gs}], \tag{12}$$

where $[\cdot||\cdot]$ presents the concatenation operation.
**FDTE Block.** We opt for a simple MLP as the Feature Dimension Transformation and Enhancement (FDTE) block. Initially, the dimensionality of $F_{CGNS}$ is reduced from $2S$ to $S$ along the channel dimension. Subsequently, the transformed $F_{CGNS}$ is further processed by a batch normalization (BN) and a ReLU activation function. This enhances the representational capacity of $F_{CGNS}$, making it more discriminative.

---

[1]Readers may be more accustomed to using the normalized Laplacian matrix $L_{sym} = I - D^{-1/2}\tilde{A}D^{-1/2}$. However, substituting $L_{sym} = I - D^{-1/2}\tilde{A}D^{-1/2}$ with $L_{sym} = D^{-1/2}\tilde{A}D^{-1/2}$ would simplify the subsequent discussions and derivations, without altering the eigenvectors. Therefore, opting to use $L_{sym} = D^{-1/2}\tilde{A}D^{-1/2}$ is feasible.

**Table 1: Evaluation on the outdoor dataset with SIFT, RootSIFT and SuperPoint for relative pose estimation. mAP5° and mAP20° are reported. The best result and the second-best result in each column are respectively bolded and underlined.**

| Method | References | Size (MB) | SIFT (%) | | RootSIFT (%) | | SuperPoint (%) | |
|---|---|---|---|---|---|---|---|---|
| | | | mAP5° | mAP20° | mAP5° | mAP20° | mAP5° | mAP20° |
| CNe [32] | CVPR2018 | **0.39** | 23.55 | 52.44 | 23.85 | 52.64 | 24.83 | 52.70 |
| OA-Net++ [55] | ICCV2019 | 2.47 | 39.33 | 66.85 | 41.25 | 69.91 | 32.18 | 66.81 |
| CL-Net [58] | ICCV2021 | 1.27 | 53.10 | 70.15 | 50.43 | 79.46 | 38.99 | 68.96 |
| MS$^2$DG-Net [11] | CVPR2022 | 2.61 | 49.13 | 70.52 | 47.20 | 75.83 | 37.38 | 68.08 |
| U-Match [21] | IJCAI2023 | 7.76 | 60.53 | 80.37 | 60.18 | 80.18 | 45.72 | 71.39 |
| NCMNet [22] | CVPR2023 | 4.77 | 63.43 | 79.82 | 64.05 | 80.59 | 48.20 | 73.40 |
| TrGa | - | 2.58 | **70.00** | **85.72** | **71.40** | **86.34** | **52.80** | **77.53** |
| C-TrGa | - | 2.59 | 66.03 | 82.37 | 65.28 | 82.24 | 48.82 | 73.60 |

## 3.5 Why is the CGNS block useful?

Global graphs in the work are undirected and sparse correspondences are permutation-invariant [55], so it can be inferred from the homophilic graph definition that these global graphs are homophilic graphs. [35, 48] points out that low-frequency signals contain more effective information in homophilic graphs. Hence, low-frequency signals contain more effective information for correspondence pruning among sparse correspondences. Therefore, proving the usefulness of the CGNS block can be transformed into the problem of proving that the CGNS block is equivalent to a low-pass filter. That is, we only need to study the properties of the frequency function $p(\lambda)$ corresponding to a normalized Laplacian matrix $L_{sym} = D^{-1/2}\tilde{A}D^{-1/2}$:

$$
\begin{aligned}
L_{sym} &= D^{-1/2}\tilde{A}D^{-1/2}, \\
&= D^{-1/2}(D - L)D^{-1/2}, \\
&= I - D^{-1/2}LD^{-1/2}, \\
&= I - L_s,
\end{aligned}
\tag{13}
$$

where the definition of the Laplacian matrix is $L = D - \tilde{A}$, so $\tilde{A} = D - L$.

Since $L_s$ can be orthogonally diagonalized, we assume $L_s = V\Lambda V^T$ and $\lambda_i$ is the eigenvalue of $L_s$, which can prove that $\lambda_i \in [0, 2)$ in [48]. Hence, Eq. (13) can be rewritten as:

$$
L_{sym} = I - V\Lambda V^\mathsf{T} = V(1 - \Lambda)V^\mathsf{T}
\tag{14}
$$

Obviously, its frequency response function is $p(\lambda) = 1 - \lambda_i \in (-1, 1]$, which is a linearly contracted function, so it can serve as a low-pass filter for graph-based data.

## 3.6 Loss Function.

Following [11, 32, 55], we select a hybrid loss function as the training objective:

$$
L = L_{cls} + \beta L_{reg}(E, \hat{E})
\tag{15}
$$

where $L_{cls}(\cdot, \cdot)$ denotes a binary cross-entropy classification loss; $\beta$ is a hyper-parameter to balance these two terms. $L_{reg}(\cdot, \cdot)$ is a geometry loss, which can be written as:

$$
L_{reg}(E, \hat{E}) = \frac{(v'^T \hat{E}v)^2}{\|Ev\|_{[1]}^2 + \|Ev\|_{[2]}^2 + \|E^T v'\|_{[1]}^2 + \|E^T v'\|_{[2]}^2}
\tag{16}
$$

**Table 2: Evaluation on the indoor dataset with SIFT and SuperPoint for relative pose estimation.**

| Method | Size (MB) | SIFT (%) | | SuperPoint (%) | |
|---|---|---|---|---|---|
| | | mAP5° | mAP20° | mAP5° | mAP20° |
| CNe | **0.39** | 9.36 | 24.98 | 10.21 | 25.68 |
| OA-Net++ | 2.47 | 16.39 | 39.87 | 12.12 | 39.82 |
| CL-Net | 1.27 | 17.03 | 37.54 | 14.03 | 37.48 |
| MS$^2$DG-Net | 2.61 | 17.84 | 38.46 | 16.08 | 40.02 |
| U-Match | 7.76 | 21.46 | 47.13 | 18.87 | 44.72 |
| NCMNet | 4.77 | 20.66 | 46.80 | 16.85 | 41.27 |
| TrGa | 2.58 | **23.54** | **49.75** | **20.35** | **46.12** |
| C-TrGa | 2.59 | 23.49 | 48.87 | 19.89 | 45.77 |

where $E$ and $\hat{E} = g(P, C)$ are the ground truth and estimated essential matrices, in which $g(\cdot)$ is the weighted eight-point algorithm; $v$ and $v'$ are virtual correspondence coordinates obtained by the ground truth $E$.

## 3.7 Implementation Details.

The input of TrGa is an $N \times 4$ putative correspondence set, in which $N$ is up to 2000. Channel dimension $S$ is 128. The Transformer block employs 4-head attentions. Batchsize and $\beta$ in Eq. (15) are selected as 32 and 0.5, respectively. Adam [36] optimizer has been chosen, and the learning rate is $10^{-3}$. TrGa is trained on NVIDIA GTX 3090 GPUs using a warmup strategy. Initially, the learning rate increases linearly for the first $10k$ iterations, after which it begins to decrease by a factor of 0.4 every $20k$ iterations.

## 4 EXPERIMENTS

## 4.1 Relative Pose Estimation.

Recovering relative poses requires to use the predicted inliers to accurately excavate relative position relationships (rotation and translation) between different camera aspects, which is a basic task for many advanced computer vision tasks, and needs robust matchers and appropriate local features. Therefore, the proposed TrGa, C-TrGa and baselines are evaluated with various local features under different scenes.

**Datasets.** Yahoo's YFCC100M dataset [45] and SUN3D dataset [49] are selected as outdoor and indoor scenes, respectively. In outdoor scenes, 68 sequences are regarded as training sequences and the

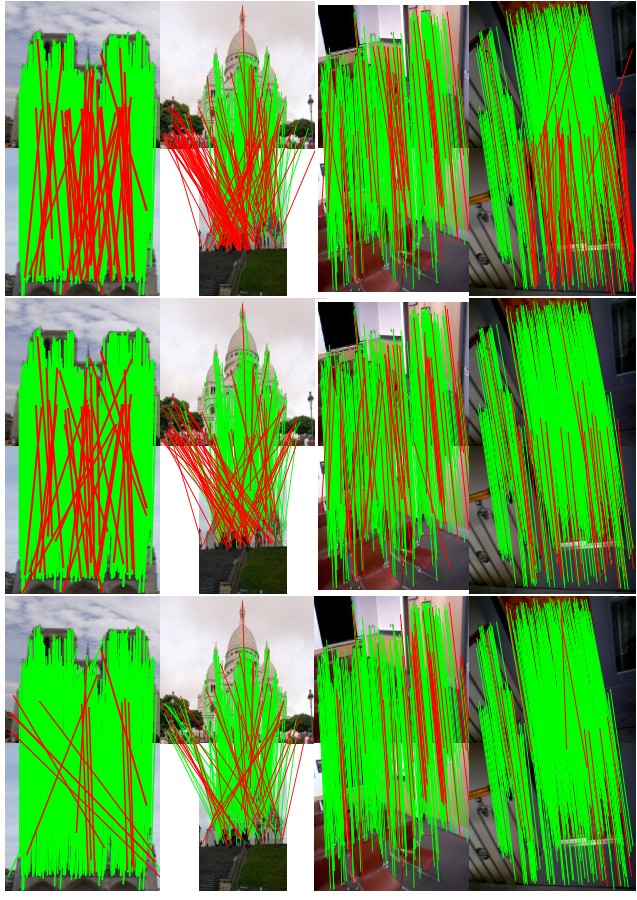

**Figure 4: Partial typical visualization results on YFCC100M and SUN3D datasets with SIFT. From top to bottom: the results of OA-Net++, MS$^2$DG-Net and the proposed TrGa.**

remaining 4 sequences are selected as testing sequences. At the same time, 239 sequences are chosen as training in the indoor dataset, and the remainders are used to test various networks.

**Evaluation Metrics.** The weighted eight-point algorithm is employed to estimate an essential matrix according to the selected true correspondences, which is decomposed to a rotation vector and a translation vector. Next, error metrics refer to angular differences between calculated rotation/translation vectors and labels, we select mAP5° and mAP20° as default metrics to evaluate different networks for relative pose estimation.

**Baselines.** We choose six learning-based networks (CNe [32], OA-Net++ [55], CL-Net [58], MS$^2$DG-Net [11], U-Match [21] and NCM-Net [22]) as baselines. CNe [32] is the first network to use deep learning technology to eliminate outliers from sparse correspondences. Afterwards, a series of networks [11, 21, 22, 55, 58] utilizing various GNNs have emerged to improve network performance.

**Outdoor Relative Pose Estimation Results.** We summary quantitative comparative experimental results of the proposed TrGa, C-TrGa and other state-of-the-art networks for relative pose estimation under outdoor scenes with three popular local features (SIFT [27], RootSIFT [1] and SuperPoint [12]) in Table 1. Notably, outdoor

**Table 3: Evaluation homography estimation on HPatches. Accuracy (ACC.) at different error thresholds is reported.**

| Method | Size (MB) | HPathces (%) | | |
|---|---|---|---|---|
| | | ACC.3px | ACC.5px | ACC.10px |
| CNe [32] | **0.39** | 38.97 | 51.55 | 65.34 |
| OA-Net++ [55] | 2.47 | 39.83 | 52.76 | 62.93 |
| CL-Net [58] | 1.27 | 43.10 | 55.69 | 68.10 |
| MS$^2$DG-Net [11] | 2.61 | 41.21 | 50.17 | 62.59 |
| U-Match [21] | 7.76 | 48.90 | 59.41 | 70.83 |
| NCMNet [22] | 4.77 | 48.79 | 58.48 | 68.00 |
| TrGa | 2.58 | **54.59** | **67.34** | **77.00** |
| C-TrGa | 2.59 | 53.09 | 66.09 | 75.89 |

scenes are challenging due to illumination changes, large-scale rotation and viewpoint changes, but the proposed TrGa performs best in all cases. In particular, the proposed TrGa improves 6.57% mAP5° on the outdoor scene with SIFT than the third best network (NCMNet), but the parameter quantity is only about half of it. Compared with OA-Net++ and MS$^2$DG-Net (with similar parameter quantities as TrGa), TrGa has improvements of 30.67% and 20.87% mAP5° on outdoor scenes with SIFT, respectively. From the first two columns of Figure 4, we can find that the proposed TrGa can remove more outliers than OA-Net++ and MS$^2$DG-Net. This is thanks to the proposed TrGa can effectively capture interaction patterns among sparse correspondence attribute features and global structural information among them in each image pair without confusion.

**Indoor Relative Pose Estimation Results.** In addition, we conduct relative pose estimation in more challenging indoor scenes with SIFT [27] and SuperPoint [12], where there are difficulties such as lack of texture information, excessive repetitive structures and so on, as shown in the right of Figure 4. Nonetheless, the proposed TrGa performs best on both SIFT and SuperPoint preprocessed datasets, as summarized in Table 2. More specifically, the mAP5° result of TrGa is 45.49% higher than OA-Net++ and 31.95% higher than MS$^2$DG-Net on indoor scenes with SIFT. It is worth mentioning that parameter quantities of OA-Net++ and MS$^2$DG-Net are similar to that of TrGa, and TrGa parameter quantity is between the two. Therefore, we visualize some typical results of the above three networks in Figure 4 and it can be seen that the image pairs processed by TrGa have fewer outliers (red lines) than others.

## 4.2 Homography Estimation.

Homography estimation is a geometric transformation between two planes, using at least 4 pairs of identical points to calculate the transformation matrix, which is the foundation for follow-up computer vision tasks. Hence, models on YFCC100M with SIFT of the proposed TrGa, C-TrGa and baselines are directly tested on HPatches benchmark [4] with Direct Linear Transform (DLT). In particular, there are 696 images and 116 scenes in HPatches benchmark, and one scene includes 1 reference image and 5 query images. Hence, there are a total of 580 image pairs, and they have significant changes in perspective or in lighting. We select SIFT to detect keypoints (up to 4000), followed by an NN strategy in each image pair. Referring to [12], homography errors below 3/5/10

**Table 4: Evaluation visual localization on Aachen Day-Night.**

| Method | Size (MB) | Day (0.25m, 2°)/(0.5m, 5°)/(1.0m, 10°) | Night |
|---|---|---|---|
| CNe [32] | **0.39** | 81.3/91.4/95.9 | 68.4/78.6/87.8 |
| OA-Net++ [55] | 2.47 | 82.3/91.9/96.5 | 71.4/79.6/90.8 |
| CL-Net [58] | 1.27 | 83.3/92.4/97.0 | 71.4/80.6/**93.9** |
| MS$^2$DG-Net [11] | 2.61 | 82.8/92.1/96.8 | 70.4/82.7/**93.9** |
| U-Match [21] | 7.76 | **85.3**/92.6/96.8 | 72.4/82.7/90.8 |
| NCMNet [22] | 4.77 | 83.1/91.4/96.8 | 69.4/80.7/89.8 |
| TrGa | 2.58 | **85.3/92.7/97.2** | **72.6/82.9/93.9** |
| C-TrGa | 2.59 | **85.3**/92.6/97.1 | 72.5/82.8/ **93.9** |

**Table 5: Quantitative comparative results of point cloud registration on the 3DMatch dataset with FPFH.**

| Method | 3DMatch RR (%↑) | RE (°↓) | TE (cm↓) |
|---|---|---|---|
| FGR [59] | 40.67 | 3.99 | 9.83 |
| SM [19] | 55.88 | 2.94 | 8.15 |
| TEASER [51] | 75.48 | 2.48 | 7.31 |
| RANSAC [14] | 73.57 | 3.55 | 10.04 |
| GC-RANSAC [5] | 67.65 | 2.33 | 6.87 |
| PointDSC [3] | 76.89 | 2.08 | 6.54 |
| TrGa | **78.66** | **2.06** | **6.53** |
| C-TrGa | 78.45 | 2.07 | **6.53** |

pixels are used to evaluate different networks. As shown in Table 3, the performance of TrGa is about 10% higher than the third best performing network (U-Match) at all thresholds. That is, our TrGa is more suitable for homography estimation than other networks, the reason of which may be that our TrGa can effectively and reasonably use attribute information of sparse correspondences and global structural information among them.

## 4.3 Visual Localization.

The purpose of visual localization is to estimate the 6-degree of freedom (DOF) relative pose, including 3-DOF for rotation and another 3-DOF for translation, on a given query image based on the corresponding 3D scene model, which is important for subsequent computer vision tasks. Hence, we compare the proposed TrGa, C-TrGa, and other comparative networks on this task. Specifically, we integrate all networks into the official HLoc [39]. In addition, we select Aachen Day-Night [40] to test different networks, which includes 922 query images (824 daytime and 98 nighttime) and 4328 reference ones, and each image of them is firstly extracted up to 4096 keypoints with SIFT followed by an NN strategy. Next, a SfM model is used to triangulate from day-time images with known poses, and register nighttime query images with 2D-2D matches gained from those correspondence learning networks and COLMAP [41]. We choose the percentage of correctly localized queries at specific distances and orientation thresholds as the evaluation matrix. From Table 4, it can be seen that the proposed TrGa performance is the best under all conditions, which can demonstrate that TrGa is suitable for visual localization.

**Table 6: Generalization ability test on YFCC100M, SUN3D and PhotoTourism (PT) with different feature extractors, including RootSIFT, SuperPoint (SP) and SIFT. mAP5° is reported.**

| Method | YFCC100M (%) SIFT | RootSIFT | SUN3D (%) SIFT | SP | PT (%) SIFT |
|---|---|---|---|---|---|
| CNe [32] | 29.25 | 30.85 | 1.03 | 2.09 | 20.17 |
| OA-Net++ [55] | 39.30 | 40.03 | 2.87 | 3.05 | 40.39 |
| CL-Net [58] | 53.35 | 53.95 | 2.88 | 2.48 | 45.54 |
| MS$^2$DG-Net [11] | 49.13 | 50.25 | 3.60 | 3.28 | 45.53 |
| U-Match [21] | 60.53 | 60.46 | 6.76 | 2.42 | 54.43 |
| NCMNet [22] | 63.43 | 64.83 | 5.46 | 2.57 | 54.73 |
| TrGa | **70.00** | **68.33** | **8.40** | **4.59** | 62.22 |
| C-TrGa | 66.03 | 66.55 | 7.46 | 3.70 | **62.57** |

## 4.4 Point Cloud Registration.

We also perform the proposed TrGa, C-TrGa, and other algorithms to complete point cloud registration on 3DMatch [54] dataset preprocessed by the traditional FPFH [38] descriptor, the division of which is the same as [8, 9], to prove the generalization and robustness of our TrGa. Notably, we choose five traditional methods (FGR [59], SM [19], TEASER [51], RANSAC [14] and GC-RANSAC [5]), and iterate RANSAC and GC-RANSAC 100$k$ times respectively. The chosen PointDSC [3] is a network proposed for point cloud registration. Specifically, we integrate TrGa and C-TrGa into the official PointDSC [3], and choose three evaluation metrics. 1) Rotation Error (RE), 2) Translation Error (TE), 3) Registration Recall (RR). Following PointDSC [3], we also define that a successful registration result, whose TE and RE are less than 30cm and 15°, respectively. The proposed TrGa and C-TrGa perform best in Table 5, which can prove that they are qualified with point cloud registration.

## 4.5 Understanding TrGa.

**Generalization Ability.** Two-view correspondence pruning networks exhibit poor generalization across different descriptors and scenes. For instance, a model trained on outdoor scenes may perform poorly in indoor scenes. Specifically, we pre-train all models on the YFCC100M dataset with SIFT and subsequently evaluate their performance on various datasets (YFCC100M, SUN3D, and PhotoTourism [16]) with different feature extractors (SIFT, SuperPoint, and RootSIFT). Notably, PhotoTourism contains a large number of tourists. As demonstrated in Table 6, the performance of both TrGa and C-TrGa consistently ranks in the top two positions across all scenarios. That is because the proposed TrGa and C-TrGa can effectively use the attribute information of sparse correspondences and the global structural information among them without confusion.
**Efficiency.** The theoretical time complexity of each standard Transformer block in TrGa and each C-Transformer block in C-TrGa is denoted as $k_s$ and $k_c$, which can be written as:

$$k_s = O(N^2 \times H) + O(N \times d^2), \quad (17)$$

$$k_c = O(S^2 \times H) + O(N \times d^2), \quad (18)$$

where $N$, $H$, $S$, and $d$ stand for the number of correspondences, attention heads, channels dimensionality, and hidden state dimensionality, respectively. $O(N \times d^2)$ corresponds to the complexity

**Table 7: Efficiency evaluation. mAP5°, the average runtime (ART, unit: ms) of each image pair on YFCC100M with SIFT and parameter size (size, MB) of different networks are reported.**

| Method | CNe [32] | OA-Net++ [55] | CL-Net [58] | MS$^2$DG-Net [11] | U-Match [21] | NCMNet [22] | TrGa | C-TrGa |
|---|---|---|---|---|---|---|---|---|
| mAP5° (↑) | 23.55 | 39.33 | 53.10 | 49.13 | 60.53 | 63.43 | **70.00** | 66.03 |
| ART (ms↓) | **28.78** | 51.36 | 58.45 | 55.62 | 81.03 | 74.28 | 61.25 | 48.75 |
| Size (MB↓) | **0.39** | 2.47 | 1.27 | 2.61 | 7.76 | 4.77 | 2.58 | 2.59 |

**Table 8: Evaluation on outdoor scenes with SIFT for relative pose estimation. $H$, $L$, and $M$ are the numbers of Transformer blocks, attention heads, and CGNS blocks, respectively.**

| mAP5°(%) | L=5 | L=5, M=1 | L=5, M=2 | L=6 | L=6, M=1 |
|---|---|---|---|---|---|
| H=1 | 54.30 | - | - | - | - |
| H=2 | 59.33 | - | - | - | - |
| H=4 | 64.50 | **70.00** | 67.95 | 67.15 | 68.03 |
| Size (MB) | **2.42** | 2.58 | 2.75 | 2.91 | 2.98 |

of the Feedforward, which remains the same for both TrGa and C-TrGa, so it can be omitted for convenience in future writings.

$$k_s = O(N^2 \times H), \tag{19}$$

$$k_c = O(S^2 \times H), \tag{20}$$

where $N$ is up to 2000 and $S$ is chosen as 128. Because $S$ is much smaller than $N$, the theoretical time complexity of C-TrGa is also smaller than that of TrGa. From Table 7, it can be observed that our TrGa and C-TrGa respectively achieve the top two performances with acceptable parameters. Meanwhile, C-TrGa ranks second in ART and outperforms CNe [32], which ranks first in ART, by 180.38% in performance.

## 4.6 Ablation Studies.

**Relationship among $H$, $L$ and $M$.** From the $1^{st}$ column of Table 8, it can be seen that attention head number $H$ in Transformers has increased from 1 to 4, and the effectiveness of the model is constantly increasing. Hence, we choose Transformers with $H$=4. Comparing with the $1^{st}$, $2^{nd}$ and $3^{rd}$ columns of the $3^{rd}$ row, we can find that the model performs best when we only use one CGNS block. If we use multiple CGNS blocks, features will tend to be consistent and feature distinguishability will become poor, which makes feature learning more difficult. [20] summarizes this phenomenon as the over-smooth problem of multi-layer graphs. From Table 8, we can know that the model with five Transformer blocks and one CGNS block is the best one.

**How to effectively perform Global Graph Construction?** The core of Global Graph Construction is the CGNS block. Therefore, the question of how to effectively perform Global Graph Construction can be transformed into how to build the CGNS block. Most of previous works directly use the core calculation formula of GCN: $F_{gcn} = L_{sym}F^T\tilde{A}$. The first step adopts $F^T\tilde{A}$ to learn attribute information of sparse correspondences. The second step utilizes $L_{sym}(F^T\tilde{A})$ to obtain global structural information, which already incorporates sparse correspondence attribute information. However, we discover that this approach leads to confusion between the sparse correspondence attribute features and their global

**Table 9: Evaluation Node, Structure, GCN, Add and CGNS operations in the Global Graph Construction.**

| | Node | Structure | GCN | Add | CGNS |
|---|---|---|---|---|---|
| mAP5°(%) | 68.40 | 69.23 | 68.15 | 68.03 | **70.00** |
| mAP20°(%) | 85.28 | 85.57 | 84.99 | 84.54 | **85.72** |

**Table 10: Ablation studies about network compositions on outdoor scenes with SIFT for relative pose estimation. TrGa (MLPs): replacing all Transformers inside TrGa with MLPs.**

| | mAP5° | mAP20° | Size (MB) |
|---|---|---|---|
| TrGa (MLPs) | 46.45 | 71.06 | 2.60 |
| TrGa (Transformers) | **70.00** | **85.72** | **2.58** |

structural information. (See Table 9.) Therefore, we firstly employ the proposed CGNS to separately capture the interaction patterns among sparse correspondence attribute features and the global structural information among them, which can prevent their confusion. Then, the FDTE block is applied for dimension transformation and feature augmentation. Additionally, we also try to perform only attribute information of sparse correspondences obtained by $F_T \times \tilde{A}$ (Node), only global structural information obtained by $L_{sym} \times (F_T)^T$ (Structure), simply adding up them (Add) and the original GCN to complete the task, respectively. From Table 9, it can be seen that the proposed CGNS block achieves the most favorable performance.

**MLPs vs. Transformers.** We replace Transformers with MLPs of similar parameter quantities. From Table 10, it can be observed that TrGa (Transformers) outperforms TrGa (MLPs) by a significant margin. For instance, on mAP5°, TrGa (Transformers) outperforms TrGa (MLPs) by 50.70%. This indicates that Transformers are more effective at feature extraction, because Transformers can naturally capture global context information, whereas MLPs cannot.

## 5 CONCLUSION

In this work, we devise a two-view correspondence pruning network named TrGa and a faster variant C-TrGa, the core of which consists of a Correspondence Feature Extractor (CFE) and a Global Graph Construction (GGC). CFE is composed of several complete Transformer structures, which effectively capture the global context information among sparse correspondences. The CGNS block, serving as the core of GGC, is capable of effectively learning the attribute information of sparse correspondences and the global structural information among them without confusion. Extensive experiments demonstrate that the proposed TrGa and C-TrGa outperform state-of-the-art networks in various tasks.

# ACKNOWLEDGMENTS

This work was partially supported by the National Natural Science Foundation of China under Grant (61925204), in part by the National Natural Science Foundation of China under Grant (62172226), and in part by the 2021 Jiangsu Shuangchuang (Mass Innovation and Entrepreneurship) Talent Program (JSSCBS20210200). Luanyuan Dai acknowledges the support of the China Scholarship Council program (Project ID: 202306840103).

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
