# OpenReview forum: "TrGa: Reconsidering the Application of Graph Neural Networks in Two-View Correspondence Pruning"
_acmmm.org/ACMMM/2024/Conference — MM2024 Poster_

### Official Review · Reviewer_VMxb · 2024-05-13

**Rating:** 4
**Confidence:** 3

**Summary:**

## **1. Summary.**
This work proposes the TrGa method to address the issues of insufficient MLP context awareness and confusion between attribute characteristics and global graphs in two-view correspondence pruning learning.

**Strengths:**

## **2. Strengths.**
i) This work has a reasonable motivation and a clear starting point.

ii) A large number of experiments have demonstrated the effectiveness of the proposed method.

iii) Open source code has promoted the development of this field.

**Limitations:**

## **3. Some minor suggestions.**
After carefully reading the entire paper, it can be seen that it has been carefully prepared. I only have a few small suggestions to provide to the author to improve the quality of the paper.

**About related work:**  In the relevant work, the authors should add the shortcomings of each subfield and the areas for improvement in this work.

**About methods:** The authors should provide an algorithm process and analyze its algorithm complexity. The loss function should also be further explained to enhance the readability of the paper.

**About experiments:** The specific details of the datasets and evaluation indicators should be provided.

**Other small details:** The future work should also be further provided to enable readers to grasp the direction of this method in the future.

**Overall, this is a good work, motivation, innovation, method presentation, and experiments are all carefully presented. And if the authors can adopt some my suggestions to improve the quality of this paper, I am willing to increase my rating.**

**Suitability:**

3

---

### Official Review · Reviewer_Vc5g · 2024-05-19

**Rating:** 3
**Confidence:** 3

**Summary:**

For correspondence pruning, the paper propose a two-view correspondence pruning network TrGa, which contains three main modules: Transformer used to extract correspondence features; CGNS that is used to capture the interaction and the global structural information among sparse correspondences; FDTE is applied for dimension transformation and feature augmentation.

**Strengths:**

1. The paper addresses the problem of two-view correspondence pruning, which seems to be effective from the experimental results.
2. The experimental evaluation is adequate, with comparisons to many state-of-the-art methods on selected datasets, which validates the effectiveness and efficiency of the proposed model.
3. The paper is clearly organized and code is released, making the proposed methodology more trustworthy.

**Limitations:**

1. The main work of this paper is to first replace MLP with transformer for feature extraction, and then input the obtained features into the proposed CGNS block. This contribution does not seem to be sufficient.
2. For Eq. (11), what is its specific meaning?  The author’s expression in the article is vague. Moreover, "X" represents the inner product or dot product? And why Eqs. (11) and (12) achieve the purpose mentioned by the author?
3. In Section 3.5, is L the Laplacian matrix of A and how is the second equation in Eq. (4) derived when \widetilde{A}=A+I_N?
4. Many efforts demonstrates that the adjacency matrix A can be regarded as a low-pass filter, so the author's explanation of the role of L_sym in Section 3.5 is weak. Why can't A be used directly to replace L_sym in Eq. (12)? What is the performance of replacing L_sym with A?

**Suitability:**

2

---

### Official Review · Reviewer_7Hb4 · 2024-05-25

**Rating:** 3
**Confidence:** 3

**Summary:**

This paper focuses on the application of graph neural networks in two-view correspondence pruning and develops a two-view correspondence pruning network, named TrGa. Specifically, the authors first use the transformer encoder to capture correspondence features. Then, the authors introduce a concatenation graph node and global structure (CGNS) block to separately capture the interaction patterns among sparse correspondence attribute features and the global structural information. The empirical experiments demonstrate the effectiveness of this paper.

**Strengths:**

1. The authors analyze the complexity of the proposed TrGa framework and further develop an efficient variant C-TrGa.
2. The authors provides the code for this paper.

**Limitations:**

1. In the graph construction stage, the authors take the inner product to generate the adjacency matrix and then add self-loop to the generated adjacency matrix. Since the generated adjacency matrix is a dense matrix, adding self-loop is redundant in my view. And the authors also do not verify the effect of not adding self-loops in the experiment.
2. The authors describe a general property of graph neural networks in Section 3.5, namely the low-pass filtering property. After the graph convolution operator, two connected nodes tend to be more similar. This conclusion has been widely verified by previous works. I don't understand how it is related to the insight of this paper. The authors need to explain it more clearly.
3. The generated dense graph is used to aggregate neighbor structural information. The authors claim that this strategy is a global method. In GNNs, sparse graphs are generally sampled from the generated dense graph to alleviate such problems. My concern is whether dense graph connectivity, despite capturing information from a global perspective, introduces additional noise information.

**Suitability:**

2

---

### Official Review · Reviewer_TcHW · 2024-05-25

**Rating:** 3
**Confidence:** 4

**Summary:**

In this paper, the authors proposed a two-view correspondence pruntning network, which termed TrGa. They use the transformer to capture the corrrespondence feature with global context information. And they introduce a concatenation graph node and gloal structure block to capture the interaction patterns among sparse coresondence atrribute features.

**Strengths:**

1.Experiments are extensive to show the effectiveness.

2.The methods are clearly described.

**Limitations:**

1. The description of Equation 7 is not clear. The QKV notations are not defined, and the dimension information is missing. Additionally, the transpose notation is not used properly.

2. The analysis of the ablation study is unclear. The specific experimental settings for each variable in Tables 9 and 10 are not provided. It is not clear how the effectiveness of CGNS and FDTE is demonstrated. The paper lacks the necessary information.

3. Given that there is a faster version (C-TrGa), it is unclear why the original TrGa is still retained.

4. Many recent relevant works have not been compared, which raises questions about the novelty of the results. or example, the following two papers are relevant and should be compared:

[1] Peng Y, Peng F, Liu Y, et al. Multi-View Consistency for Correspondence Pruning[C]//2023 9th International Conference on Big Data and Information Analytics (BigDIA). IEEE, 2023: 356-363.

[2] Liao T, Zhang X, Zhao L, et al. VSFormer: Visual-Spatial Fusion Transformer for Correspondence Pruning[C]//Proceedings of the AAAI Conference on Artificial Intelligence. 2024, 38(4): 3369-3377.

[3] J, Xiao G, Wang S, et al. Graph context transformation learning for progressive correspondence pruning[C]//Proceedings of the AAAI Conference on Artificial Intelligence. 2024, 38(3): 1968-1975.

5. Transformer models have already been extensively researched, so what are the specific innovations presented in this work?

[2] Liao T, Zhang X, Zhao L, et al. VSFormer: Visual-Spatial Fusion Transformer for Correspondence Pruning[C]//Proceedings of the AAAI Conference on Artificial Intelligence. 2024, 38(4): 3369-3377.

[4]Liao T, Zhang X, Xu Y, et al. SGA-Net: A sparse graph attention network for two-view correspondence learning[J]. IEEE Transactions on Circuits and Systems for Video Technology, 2023.

**Suitability:**

2

---

### Meta-Review · Area_Chair_62Jm · 2024-07-02

**Recommendation:** Accept (Poster)
**Confidence:** 4

**Metareview:**

The paper addresses a fundamental task of two-view correspondence pruning. Despite the simplicity of the proposed approach, its performance is strong. The initial concerns of the reviewers seem to be well addressed by the rebuttal. The comments from the lone reviewer who gave a "Borderline Reject" seem to be answered well by the authors (and the reviewer did not update the rating or add any comments after the rebuttal).

I'm siding with the majority opinion of the reviewers and the author rebuttal .
.